# OpenWebMath: An Open Dataset of High-Quality Mathematical Web Text

♣**Keiran Paster**,* †**Marco Dos Santos**,* °**Zhangir Azerbayev**, ♣**Jimmy Ba**

♣University of Toronto; Vector Institute for Artificial Intelligence
†University of Cambridge, °Princeton University
`keirp@cs.toronto.edu, mjad3@cam.ac.uk`

## Abstract

There is growing evidence that pretraining on high quality, carefully thought-out tokens such as code or mathematics plays an important role in improving the reasoning abilities of large language models. For example, Minerva, a PaLM model finetuned on billions of tokens of mathematical documents from arXiv and the web, reported dramatically improved performance on problems that require quantitative reasoning. However, because all known publicly released web datasets employ preprocessing that does not faithfully preserve mathematical notation, the benefits of large scale training on quantitive web documents are unavailable to the research community. We introduce OpenWebMath, an open dataset inspired by these works containing 14.7B tokens of mathematical webpages from Common Crawl. We describe in detail our method for extracting text and LaTeX content and removing boilerplate from HTML documents, as well as our methods for quality filtering and deduplication. Additionally, we run small-scale experiments by training 1.4B parameter language models on OpenWebMath, showing that models trained on 14.7B tokens of our dataset surpass the performance of models trained on over 20x the amount of general language data. We hope that our dataset, openly released on the Hugging Face Hub, will help spur advances in the reasoning abilities of large language models.

## 1 Introduction

Advances in large language models have opened up new opportunities in numerous fields, providing a transformative shift in our approach to a wide range of complex problems (Brown et al., 2020; Raffel et al., 2020). Among these problems, mathematical reasoning has drawn the attention of several researchers in recent years, becoming both a common benchmark to judge the performance of large language models and inspiring new approaches to improve their reasoning capabilities in the hope that they will one day be able to solve complex mathematical problems. One of the biggest advancements in mathematical reasoning in recent years has been the Minerva model (Lewkowycz et al., 2022), which achieved state-of-the-art results on quantitative reasoning benchmarks such as MATH (Hendrycks et al., 2021). Minerva was trained by finetuning PaLM (Chowdhery et al., 2022) on a curated dataset consisting of billions of tokens of high quality technical content sourced from both scientific papers and the web.

Minerva and the datasets used for its training were not released publicly and the current capabilities of open-source models (e.g., Touvron et al. (2023b;c;a); Geng & Liu (2023); Biderman et al. (2023)) in quantitative reasoning lags behind. We believe that there are important research directions that can only be enabled through open-access to such models and datasets, such as work on memorization and generalization, reinforcement learning, the development of new reasoning benchmarks, and advancement in the reasoning capabilities of language models.

In our work, we produce an open alternative to the Math Web Pages dataset used to train Minerva (Lewkowycz et al., 2022). We extract documents from Common Crawl[1], applying our pipeline to

---

*Keiran and Marco created the dataset and Zhangir led model training and evaluation.
[1] `https://commoncrawl.org/`

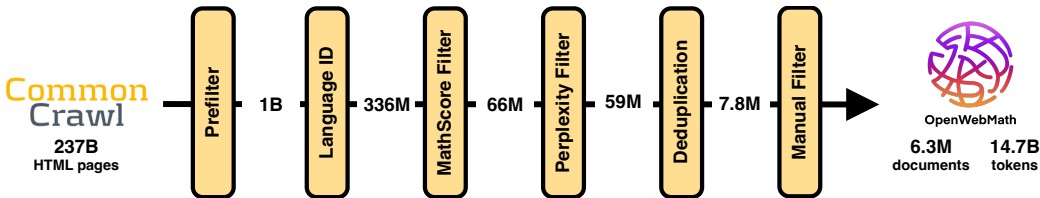

Figure 1: The pipeline for constructing OpenWebMath involves aggressive filtering so that the final dataset only contains high quality, English, and mathematical content.

extract text while preserving mathematical content in the form of LATEX equations. We then filter the documents, ensuring that only high-quality English mathematical documents are kept. Finally, we deduplicate the dataset, resulting in 14.7B tokens of high-quality mathematical content suitable for both pretraining and finetuning large language models. The key contributions of this work are as follows:

- We publically release OpenWebMath, a dataset of 14.7B tokens of high-quality mathematical web text. Our dataset can be found at https://huggingface.co/datasets/open-web-math/open-web-math on the Hugging Face Hub.

- We extensively document our pipeline, sharing our findings with the NLP community. We open-source the code needed to reproduce our results.

- We analyze the quality of OpenWebMath. First, we analyze the contents of our dataset, providing statistics on the types of webpages, subjects, and top domains. Then, we train several language models on our dataset to show that per-token, it is more effective than existing mathematical pretraining datasets, and is most effective when combined with other datasets.

## 2    RELATED WORK

### 2.1    MATHEMATICS DATASETS AND BENCHMARKS

**Mathematics datasets**    Over the past couple of years, several datasets of mathematics have been introduced. AMPS, a dataset of informal mathematics, was introduced alongside the MATH dataset (Hendrycks et al., 2021). AMPS includes more than 100,000 Khan Academy problems with step-by-step solutions in LaTeX and over 5 million problems generated using Mathematica scripts. In total, AMPS contains 23GB of problems and solutions. Another notable example is NaturalProofs (Welleck et al., 2021), which encompasses 32,000 theorem statements and proofs, 14,000 definitions, and 2,000 other types of pages (e.g. axioms, corollaries) derived from ProofWiki, the Stacks project and data from mathematics textbooks. Proof-Pile (Azerbayev et al., 2023) is a dataset of mathematical text that contains more than 14.5GB of informal mathematics texts obtained from arXiv, Stack Exchange, ProofWiki, Wikipedia, openly licensed books, and the MATH dataset. There are also many proprietary datasets for mathematics. WebMath is a large-scale dataset mentioned by OpenAI researchers (Polu & Sutskever, 2020) that contains a 35B token mix of content from Github, arXiv, and Math StackExchange, adding up to 35GB of informal mathematics. MathMix is another OpenAI dataset used to finetune GPT-4 (Lightman et al., 2023) that contains 1B high quality mathematical tokens containing both natural and synthetic data. The proprietary web dataset used to train Minerva, called Math Web Pages (Lewkowycz et al., 2022), was compiled by collecting 17.5B tokens from web pages that contain LATEX code.

**Mathematics benchmarks**    Several popular benchmarks have been used by researchers to assess the capabilities of language models on both formal and informal mathematics. The MATH dataset (Hendrycks et al., 2021) is comprised of 12,500 challenging competition problems in informal language. Each problem is also accompanied by a step-by-step informal proof. Answers are delimited by the \boxed environment, allowing for easier answer verification. GSM8k (Cobbe et al., 2021) is another popular multi-step informal mathematics reasoning benchmark. It contains 8,500 grade

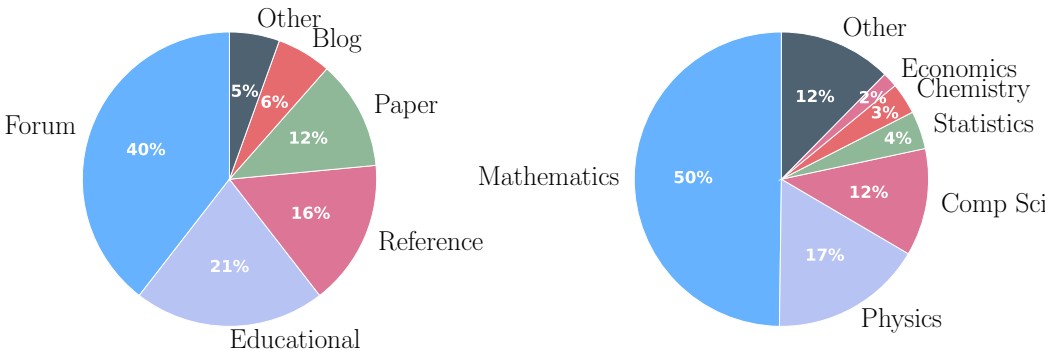

Figure 2: **Left**: The documents in OpenWebMath are sourced from forum posts, educational content, reference pages, scientific papers, blogs, and more. Most content comes from Q&A forums where users discuss how to solve problems. **Right**: The majority of the content in OpenWebMath is related to mathematics, but a large part is related to other technical subjects like Physics, Computer Science, Statistics, and more.

school math problems that are intended to be solvable by a bright middle school student. Lewkowycz et al. (2022) also introduce a benchmark based on OpenCourseWare. OCWCourses includes a set of 272 automatically-verifiable solutions at the undergraduate level, covering chemistry, information theory, differential equations, special relativity, and more. Lewkowycz et al. (2022) also evaluate on a subset of MMLU (Hendrycks et al., 2020) called MMLU-STEM, which focuses on science, technology, engineering, and mathematics.

## 2.2 WEB DATA PROCESSING PIPELINES

The pretraining of large language models requires large, diverse datasets. Data scraped from the web is one of the primary sources for such data. However, sources such as Common Crawl, which contains over 200 billion web pages, are known to have significant amounts of low-quality and duplicate content, requiring extensive filtering and deduplication to be suitable for training. Prior works such as C4 (Raffel et al., 2020), RefinedWeb (Penedo et al., 2023), CCNet (Wenzek et al., 2019), The Pile (Gao et al., 2020), and GPT-3 (Brown et al., 2020) introduce various pipelines for extracting quality data from Common Crawl for the purposes of language model training. These pipelines typically consist of three primary steps: text extraction, filtering, and deduplication.

**Text extraction** Extracting plain text from HTML files is a critical step in the creation of Common Crawl-based datasets. The easiest way to extract text from Common Crawl documents is to use the WET corresponding to each webpage, which contains pre-extracted plain text of the webpage. CCNet and C4 both use Common Crawl's WET files. However, the text extracted in WET files may contain too much boilerplate or miss out on important content such as LaTeX equations. It is also possible to extract text directly from the raw HTML found in Common Crawl WARC files. The Pile uses an open source library called jusText (Endrédy & Novák, 2013) to extract text from HTML while RefinedWeb uses a library called Trafilatura (Barbaresi, 2021). These text extraction approaches differ in terms of extraction speed, customization, and their precision and recall for removing boilerplate content.

**Filtering** The first layer of filtering often involves language identification (Wenzek et al., 2019). Language filtering is used because certain other parts of the pipeline only work for specific languages, and is often done with simple linear classifiers such as from fastText (Joulin et al., 2016). Quality filtering can be done with a combination of perplexity, classifier, and rule-based methods. CCNet uses a 5-gram Kneser-Ney language model implemented in the KenLM library (Heafield, 2011) trained on the target domain. The documents in the dataset are then sorted and filtered by their perplexity under this model. Other datasets such as the one used to train GPT-3 (Brown et al., 2020) use a classifier-based approach. This involves training a classifier on known-high-quality

documents, such as those from Wikipedia, as positive examples and unfiltered documents from Common Crawl as negative examples. The classifier scores are used to filter low-quality documents from the dataset. Finally, rule-based approaches such as those used in C4 (Raffel et al., 2020) and MassiveWeb (Rae et al., 2021) involve removing pages with certain characters, too many or too few characters, too high a proportion of symbols, or those with an abnormal average word length. OpenMathWeb uses a mixture of these three approaches.

**Deduplication** Given the periodic nature of Common Crawl snapshots and a general redundancy in web-sourced text, deduplication is an important processing step. Document-level near-deduplication (e.g., in (Brown et al., 2020; Penedo et al., 2023)) often employs MinHashLSH, an efficient algorithm for estimating the Jaccard similarity of documents. CCNet (Wenzek et al., 2019) uses paragraph-level deduplication, which can help to remove common boilerplate content found in WET text-extractions.

## 3 BUILDING OPENWEBMATH

### 3.1 OBJECTIVES

Our aim with OpenWebMath is to build a dataset of as many mathematical documents sourced from the web as possible while preserving the formatting of mathematical content such as LaTeX equations as in Lewkowycz et al. (2022). For the purposes of this work, we define a mathematical document as a document containing either core mathematical contents such as theorems, definitions, proofs, questions and answers, formal mathematics, or interdisciplinary documents featuring mathematical formulas within fields like physics, chemistry, biology, economics, and finance. We source our documents from Common Crawl, which is a large open-access crawl of the web containing petabytes of raw HTML files. Due to the high variance in the quality of documents from Common Crawl, we additionally use several methods for filtering and boilerplate reduction. Throughout the creation of OpenWebMath, we iteratively refined these methods to ensure that we do not remove too many relevant documents, optimizing for high recall whenever possible. Since we expect that OpenWebMath will be used primarily as an additional source of pretraining data for large language models, we prefer having a small percentage of non-mathematical but high quality documents in the dataset rather than removing them and potentially losing relevant mathematical content. Finally, due to the limited number of mathematical data available on the web, we use significantly more manual inspection and tuning of our processing pipeline than other web-based datasets. We document our processing choices and pipeline in the section that follows.

### 3.2 HIGH-LEVEL OVERVIEW OF THE PIPELINE

As shown in Figure 1, the processing pipeline for OpenWebMath falls into five stages. First, we apply a prefilter to all HTML documents in Common Crawl to quickly judge whether they have mathematical content, skipping those that do not before doing the extensive processing needed to extract text and equations and remove boilerplate. Second, we extract the text, including mathematical content, from the HTML documents. Third, we apply language identification filters, perplexity-based quality filtering, and a mathematical content classifier filter. Fourth, we deduplicate the dataset using SimHash (Manku et al., 2007). Finally, we manually inspect the documents gathered in the previous steps and view documents from the most popular domains by document-count and character-count, removing domains that are not high quality. We describe each of these steps in detail in the following sections.

### 3.3 PREFILTERING

Since there are over 200B HTML documents in Common Crawl, applying our processing over each document would require a significant amount of compute. To improve the efficiency of the pipeline, we first apply a stack of pre-filters optimized for high recall to reduce the number of documents that need to be processed. Our first filters check for common mathematical strings as in Lewkowycz et al. (2022), such as the presence of `tex` classes, `<math>` tags, and the word "mathjax". See Table 11 for a full list of terms. If none of these terms are present, we search for the presence of the top 100 most-popular LaTeX symbols in the text. This is done by first filtering for documents

```
This paper concerns the quantity
<img src="https://s0.wp.com/
latex.php?latex=%7BM%28x%29..."
alt="{M(x)}" />, defined as the
length of the longest
subsequence of the numbers from
```

```
Suppose I have a smooth map
[tex]f\colon \mathbb{R}^3
\longrightarrow S^2[/tex]. If I
identify [tex]\mathbb{R}^3[/tex]
with [tex]U_S = S^3 - \
{(0,0,1)\}[/tex] via
stereographic projection
```

```
<math>
  <semantics>
    ...
    <annotation ...>
      {\displaystyle \mathrm {MA}
      ={\frac{f_{O}}{f_{E}}}}
    </annotation>
  </semantics>
</math>
```

| Image Equations | Delimited Math | Special Tags |

Figure 3: LATEX formulas can be embedded in HTML documents in many ways, including in images, within arbitrary delimiters, and within special tags. Most common text-extraction pipelines do not extract LATEX code properly.

containing a backslash command using a simple regular expression and then searching specifically for these LATEX symbols in the plain text from the HTML document. If none of these symbols are found, we run the plain text through our *MathScore* classifier (see section 3.5.1) and keep documents that exceed a confidence threshold of 0.8. By tuning these filters and using hierarchical layers of progressively more accurate but more expensive filters, we were able to reduce the compute needed to process the dataset by several times while retaining a high recall of relevant documents.

## 3.4 TEXT EXTRACTION

In contrast with prior works that extract text from Common Crawl such as C4 (Collins et al., 2023), The Pile (Gao et al., 2020), and RefinedWeb (Penedo et al., 2023), we chose to make a mostly custom pipeline for extracting the main content from HTML documents. This is because we found that while other tools get decent performance on average over many documents on the internet, they do not work optimally on many of the most common sources of mathematical content on the web. We instead opted to build on top of Resiliparse (Bevendorff et al., 2018; 2021), a fast and efficient library built in Cython that includes performant tools for parsing HTML pages, processing their DOMs, and extracting the main content. As shown in Table 8 in the appendix, Resiliparse is significantly more efficient than alternative libraries such as jusText. Another notable part of our text extraction pipeline is that we randomize the parameters of the extraction to add diversity to the dataset. This includes randomizing whether we use a plain text or Markdown format for the documents and randomizing the amount of boilerplate terms required to trigger a line being removed.

Our text extraction pipeline consists of four stages: LATEX extraction, text extraction, DOM processing, and line processing.

**LATEX Extraction** Lewkowycz et al. (2022) employ a relatively simple LATEX extraction pipeline that extracts equations from `<script type="math/latex">`, `<script type="math/asciimath">`, and `<math>` blocks with `<annotation encoding="application/x-tex">` blocks within them and replaces these tags with the extracted equations. When we applied these filters to documents from Common Crawl, we noticed an extremely low number of these tags compared to what was reported. We suspect that this is due to a difference between the HTML files available within Google (Lewkowycz et al., 2022) and those available on Common Crawl. The majority of the LATEX on the internet is written using MathJax, where developers write equations delimited by dollar signs or other delimiters in their HTML pages and then the included javascript code replaces these equations with properly rendered LATEX equations within the above script tags when the page is loaded. HTML documents on Common Crawl do not include the changes to the HTML that result from running javascript, requiring that we instead extract the LATEX equations by finding delimiters ourselves. This is a significant challenge since we need to detect whether the page contains the required MathJax javascript code, which delimiters were chosen by the user to denote equations, and then match and extract the equations from the text on the page. See Appendix D for a more detailed discussion.

In order to extract MathJax, we first determine whether the page is importing the MathJax javascript code by searching for the word MathJax on the page. If it is not found, we additionally search for common LATEX symbols, and if they are found, we treat the page as though it is running MathJax. We use regular expressions to search for code that calls the configuration function for MathJax to

extract the delimiters used for equations. We add these delimiters to an extensive list of default delimiters and treat any content between these delimiters as LaTeX equations.

In addition to extracting equations from MathJax, we found several more ways that LaTeX is encoded on the internet. These methods were discovered by filtering small portions of Common Crawl for documents that contain `\frac`, one of the most popular LaTeX commands, and making sure that our processing code supports all the different ways that math could be encoded. We found that LaTeX on the internet is encoded in the following ways:

1. `equation` and `align` environments.

2. The `alttext` of elements with special classes like `tex`.

3. Images from domains like `latex.codecogs.com` often include equations encoded in the URL.

4. Special wordpress plugins.

5. `<math>` tags with `<annotation encoding="application/x-tex">` blocks within them.

6. `<math>` tags with MathML content. We use a style sheet to convert these equations into LaTeX.

7. MathJax equations encoded in the text of the page.

The relative frequencies of the different ways math is encoded can be found in Table 9 in the appendix.

**DOM Processing**    After extracting the LaTeX equations from the HTML, we do several processing steps on the DOM-tree of the HTML document. This includes removing invisible elements based on their styles, removing buttons and link clusters, annotating code, tables, and headers, and removing known problematic elements based on class or ID.

**Text Extraction**    We use the `extract_plain_text(main_content=True)` method in Resiliparse (Bevendorff et al., 2018) to extract the main content text from the DOM following several preprocessing steps to get around common issues with their specific implementation that cause it to be overly sensitive when removing boilerplate.

**Line Processing**    After extracting the plain text on the page using Resiliparse, we apply our own processing to remove boilerplate lines based on an iteratively-refined set of common boilerplate phrases, remove empty headers, and escape dollar signs that are not part of LaTeX equations.

### 3.5    FILTERING

We apply filtering with the goal of removing non-English documents (since our filters pipeline is optimized for English), removing documents that are not mathematical, and removing low-quality documents that would be harmful to train a language model on. We apply the following filters in order:

1. We use a FastText language identification model (Joulin et al., 2016) to remove documents that are not in English.

2. We use our *MathScore* classifier (see section 3.5.1) to get a probability that the document is mathematical. If our previous extraction step found LaTeX equations, we keep documents with a probability of over 0.17. If no LaTeX equations were found, we keep documents with a probability of over 0.8.

3. We use a KenLM language model (Heafield, 2011) trained on ProofPile (Azerbayev et al., 2023) to get a perplexity score for each document. We remove documents with a perplexity score of more than 15,000.

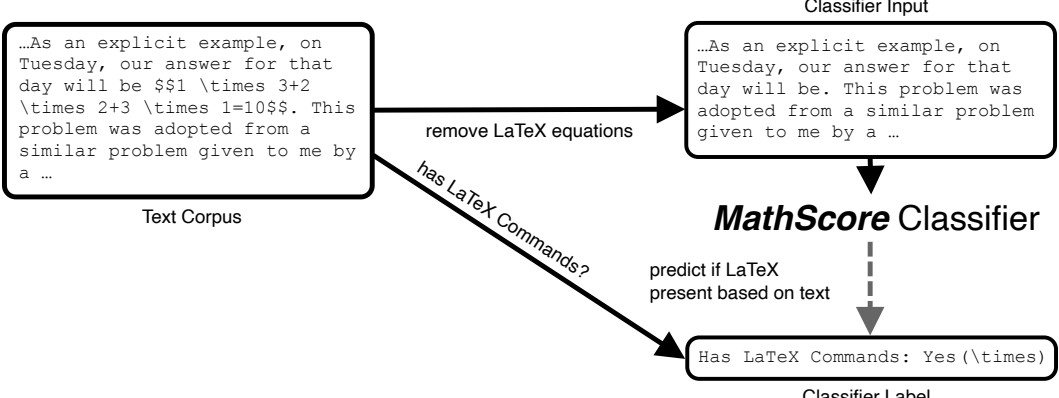

Figure 4: The MathScore classifier used in filtering OpenWebMath is trained to predict whether a text has any of the most popular LaTeX commands based only on surrounding words. This lets us include documents on the web that do not include extractable LaTeX but still contain technical content.

### 3.5.1 MATH SCORE

During our filtering process, we train a model to predict the probability a document is mathematical, which we call *MathScore*. We first gather a dataset of hundreds of thousands documents extracted from our pipeline from an early stage of the project, and label them depending on whether they contain one of the top-100 most common LaTeX commands. We then remove any LaTeX code from the documents and train a classifier to predict whether the documents contain one of these common LaTeX commands. The training process for *MathScore* is depicted in Figure 4. Since we remove all LaTeX code from the features fed into the model, the model needs to learn the words and phrases most commonly associated with LaTeX content. We use FastText (Joulin et al., 2016) to train this model, and find based on manual inspection that content with a score of under 0.2 is very unlikely to contain useful mathematical content.

### 3.6 DEDUPLICATION

Due to the large amount of duplicate documents in Common Crawl, we apply a deduplication step to remove near-duplicate documents. We use the SimHash implementation from text-dedup (Mou et al., 2023) to deduplicate the dataset using a threshold of 0.7. We find that this threshold is high enough to remove most duplicate documents even if they have slight differences in their texts.

### 3.7 MANUAL INSPECTION

Finally, we manually inspect the top domains by document count, the top domains by character count, and the longest documents in the dataset to ensure that the documents are high quality. We remove domains that are not high quality or clearly not mathematical by adding domains to a black-list and adding domain filters such as removing user profile pages, abstract-hosting websites as in Lewkowycz et al. (2022), and removing search result pages.

## 4 DATASET ANALYSIS

**Token count** At 14.7B tokens, OpenWebMath is just below the size of Minerva's Math Web Pages (17.5B tokens) Lewkowycz et al. (2022) and significantly larger than the web part of any other dataset. OpenWebMath has around the same number of LLaMA tokens as ProofPile (14.2B) (Azerbayev et al., 2023), but we note that there is very little overlap between the two datasets. Table 1 shows the token compositions of ProofPile (Azerbayev et al., 2023) and OpenWebMath in relation to the entire Minerva (Lewkowycz et al., 2022) training dataset. OpenWebMath brings a large number of new mathematical tokens that were previously unavailable to the open-source community. Due to differences in data curation strategies, it is hard to compare these datasets other than by training

models on them. Since not much is known about how to properly filter a dataset, we opted to keep as much relevant content as possible. However, future work could explore filtering OpenWebMath more aggressively to further improve its quality.

**Data Composition** We measured the distribution of domains in OpenWebMath both by document and by character count. Table 2 and Table 3 show the top twenty most common domains by document and by character count respectively. The most common sources of data tend to be discussion forums, blog posts, and scientific papers. We find that the distribution of characters in the dataset is distributed over 131,206 domains, with 46% of the characters appearing in the top 100 domains.

In order to get a sense of the types of documents found in the dataset, we analyzed 100,000 randomly sampled documents. First, we created embeddings of this data using `all-MiniLM-L12-v2` (Wang et al., 2020) in SentenceTransformers (Reimers & Gurevych, 2019). Then, we clustered these embeddings using $k$-Means with $k = 128$. Finally, we took the five closest documents to each cluster center and asked `gpt-3.5-turbo` (https://platform.openai.com/docs/api-reference) to classify each cluster as Math, Physics, Statistics, Chemistry, Economics, Computer Science, or Other. We then aggregated these statistics, using the size of each cluster to get an estimate of the final number of documents in each category. We note several potential issues with this methodology, including inaccuracies stemming from using an LLM for classification, and the potential that not every document within a cluster belongs to the predicted category. Figure 2 shows the results of this analysis. The majority of the documents in the dataset are directly related to mathematics, while the rest are spread out throughout physics, computer science, statistics, chemistry, and economics, with 12% of documents not falling neatly into any of these categories.

We also used GPT to analyze the types of websites found in OpenWebMath. To do this, we took a sample of 200 documents and asked `gpt-3.5-turbo` to classify each as a Forum, Paper, Blog, Reference, Educational, Reference, or other. We also gave the document URL as a feature, since

| Dataset | Tokens (in billions) | | | Openly Available |
|---|---|---|---|---|
| | arXiv | Formal math | Web Pages | |
| Minerva (Google) | 21.0 | 0.0 | 17.5 | ✗ |
| ProofPile | 13.0 | 0.2 | 0.8 | ✓ |
| **OpenWebMath** | **0.0** | **0.0** | **14.7** | ✓ |

Table 1: We used the LLaMA tokenizer to count ProofPile and OpenWebMath tokens. A small amount of ProofPile tokens related to textbooks are not included in this table.

| Domain | # Documents | % Documents |
|---|---|---|
| stackexchange.com | 1,136,407 | 17.99% |
| physicsforums.com | 300,044 | 4.75% |
| mathhelpforum.com | 170,721 | 2.70% |
| socratic.org | 133,983 | 2.12% |
| mathoverflow.net | 120,755 | 1.91% |
| gradesaver.com | 96,100 | 1.52% |
| zbmath.org | 91,939 | 1.46% |
| wordpress.com | 87,876 | 1.39% |
| github.io | 81,125 | 1.28% |
| brilliant.org | 68,573 | 1.09% |
| gamedev.net | 50,560 | 0.80% |
| openstudy.com | 49,041 | 0.78% |
| gmatclub.com | 48,812 | 0.77% |
| blogspot.com | 48,036 | 0.76% |
| wikipedia.org | 46,606 | 0.74% |
| ac.uk | 41,342 | 0.65% |
| nature.com | 37,403 | 0.59% |
| aimsciences.org | 36,368 | 0.58% |
| libretexts.org | 32,216 | 0.51% |
| readthedocs.io | 31,455 | 0.50% |

Table 2: Most Common Domains by Document Count.

| Domain | # Characters | % Characters |
|---|---|---|
| stackexchange.com | 4,655,132,784 | 9.55% |
| nature.com | 1,529,935,838 | 3.14% |
| wordpress.com | 1,294,166,938 | 2.66% |
| physicsforums.com | 1,160,137,919 | 2.38% |
| github.io | 725,689,722 | 1.49% |
| zbmath.org | 620,019,503 | 1.27% |
| wikipedia.org | 618,024,754 | 1.27% |
| groundai.com | 545,214,990 | 1.12% |
| blogspot.com | 520,392,333 | 1.07% |
| mathoverflow.net | 499,102,560 | 1.02% |
| gmatclub.com | 442,611,169 | 0.91% |
| gamedev.net | 426,478,461 | 0.88% |
| ac.uk | 402,111,665 | 0.83% |
| aimsciences.org | 344,716,386 | 0.71% |
| mathhelpforum.com | 319,215,756 | 0.65% |
| deepai.org | 313,512,520 | 0.64% |
| libretexts.org | 282,014,149 | 0.58% |
| readthedocs.io | 269,816,413 | 0.55% |
| tib.eu | 199,714,017 | 0.41% |
| mit.edu | 198,487,362 | 0.41% |

Table 3: Most Common Domains by Character Count.

| Training Dataset | GSM8k | MATH | | | | | | |
|---|---|---|---|---|---|---|---|---|
| | | Prealgebra | Algebra | Intermediate Algebra | Counting & Probability | Number Theory | Precalculus | Geometry |
| **The Pile** (14.7B tokens) | 2.2032 | 1.9127 | 1.9751 | 1.8420 | 1.8193 | 1.9227 | 1.6847 | 1.9499 |
| **ProofPile** (14.7B tokens) | 2.2350 | 1.7370 | 1.7214 | 1.5739 | 1.6462 | 1.7291 | 1.4838 | 1.7229 |
| **OpenWebMath** (14.7B tokens) | 1.9075 | 1.6285 | 1.6503 | 1.5949 | 1.6002 | 1.6894 | 1.4542 | 1.5748 |
| **Mixture** (14.7B tokens) | **1.8968** | **1.6055** | **1.6190** | **1.5301** | **1.5719** | **1.6607** | **1.4119** | **1.5599** |
| **The Pile** (300B tokens; Pythia 1.4B) | 1.9430 | 1.7117 | 1.7560 | 1.6358 | 1.6359 | 1.7460 | 1.5191 | 1.7252 |

Table 4: We trained 1.4B parameter models for 14.7B tokens on various datasets and measured their perplexity on different mathematics benchmarks. Both OpenWebMath and a 50/50 mixture of ProofPile Azerbayev et al. (2023) and OpenWebMath perform well - outperforming Pythia 1.4B (Biderman et al., 2023) trained on 300B tokens of The Pile (Gao et al., 2020).

| Training Dataset | MATH Algebra-Easy | MATH Algebra-Easy maj@16 | LILA multiarith |
|---|---|---|---|
| **The Pile** (14.7B tokens) | 2.81% | 3.93% | 9.77% |
| **ProofPile** (14.7B tokens) | 2.81% | 3.93% | 8.04% |
| **OpenWebMath** (14.7B tokens) | **5.62%** | 9.55% | **16.67%** |
| **Mixture** (14.7B tokens) | 5.06% | **10.11%** | 13.22% |
| **The Pile** (300B tokens; Pythia 1.4B) | 3.93% | 5.62% | **21.80%** |

Table 5: Accuracy on Different Math Benchmarks.

we found GPT is often able to judge the topic from the URL alone. We validated our analysis by asking GPT to do this classification on the top 100 domain names and got similar results. Figure 2 shows the results. The highest proportion of documents are forum pages, where users ask and answer questions related to mathematical subjects. There is also a large proportion of educational and reference content.

**Downstream Performance**    We ran experiments to find out how our dataset compares to other language modeling datasets. We compare models trained on OpenWebMath for a single epoch (14.7B tokens) with models trained for the same number of tokens on The Pile (Gao et al., 2020), a general langauge modeling dataset, and ProofPile (Azerbayev et al., 2023), a dataset of both formal and informal mathematics. We also train a 50/50 mixture of ProofPile and OpenWebMath to evaluate the performance of OpenWebMath when included in a mixture of other datasets, as would be common in practice.

We train randomly initialized models with the same architecture as Pythia 1.4B (Biderman et al., 2023). We use a batch size of 1M tokens and the same hyperparameters as Pythia otherwise. These models are evaluated on a collection of mathematics benchmarks which show signal on models of this size. This includes the subset of level-1 algebra questions from MATH, LILA-multiarith to test coding ability, and GSM8k and MATH perplexities, which scale more smoothly than accuracies. We also compare to Pythia 1.4B (Biderman et al., 2023), which was trained on 300B tokens of The Pile (Gao et al., 2020) with the same architecture.

Table 4 shows the results for our perplexity evaluations. There is a clear performance lead for models trained with OpenWebMath and the mixture seems to perform best. We suspect that since OpenWeb-Math contains many more examples of step-by-step thinking and discussion of basic mathematics relative to ProofPile (Azerbayev et al., 2023) (which primarily contains arXiv data), models trained on it excel at mathematical problem solving. Despite Pythia being trained on over 20x the number of tokens, the performance of our models on the perplexity benchmarks far exceeds its performance, showing the potential of domain-specific models for mathematics. Similarly, Table 5 shows the performance of the models on MATH-Algebra-Easy and LILA-multiarith (Mishra et al., 2022). OpenWebMath models outperform models that were not trained on it by a significant margin.

## 5 CONCLUSION

In this paper, we describe OpenWebMath, an open dataset of 14.7B high quality mathematical documents from the web. We extensively document our pipeline, including several novel methodologies

for extracting LaTeX formulas, reducing boilerplate, and filtering the dataset. OpenWebMath consists of high quality Q&A forum posts, educational documents, blogs, and more spread across mathematics, physics, computer science, and other technical domains. We also train several models on Open-WebMath and other language modeling datasets to compare the downstream performance achievable by training on our dataset. Notably, we find that models trained on OpenWebMath outperform models trained on 20x more general-domain tokens in mathematics. We hope that OpenWebMath can lead to the creation of language models with improved mathematical reasoning capabilities.

## ACKNOWLEDGEMENTS

JB is supported by NSERC Grant [2020-06904], CIFAR AI Chairs program, Google Research Scholar Program, and Amazon Research Award. KP is supported by an NSERC PGS-D award. Resources used in preparing this research were provided, in part, by the Province of Ontario, the Government of Canada through CIFAR, Fujitsu Limited, and companies sponsoring the Vector Institute for Artificial Intelligence (`www.vectorinstitute.ai/partners`). Computing resources for model training were provided by EleutherAI and Brigham Young University. We thank Finn Paster for the graphic design for the logo. We additionally thank Ziming Chen, Yuhuai Wu, Stella Biderman, Aviya Skowron, Hailey Schoelkopf, and Sean Welleck for their helpful comments.

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

| Benchmark | Problem Statements with 30-gram Overlap | Total Problems |
|---|---|---|
| MATH Test | 348 | 5000 |
| GSM8k | 2 | 1319 |
| LILA (all numerical tasks) | 3 | 1310 |

Table 6: We measured the 30-gram overlap between OpenWebMath and common test sets. We observe that there is little overlap.

## A    LIMITATIONS AND FUTURE WORK

Despite the high quality of OpenWebMath, we note several limitations and avenues for future works. First, due to the high cost of extracting data from all shards on Common Crawl, we were only able to run our pipeline once. Therefore, many of our choices are without empirical justification and we provide no ablation study. We also note that the nature of this particular type of dataset means that there are many subjective choices to be made. For instance, what counts as a mathematical document? What is a high-quality document? How do we choose the threshold for near-deduplication? For each of these, we chose several values and manually inspected a few examples to choose. Due to the cost constraints, there are also practical challenges with balancing cost with accuracy when filtering and extracting text. For instance, our prefilter reduces the number of HTML documents processed to under 1% of the documents in Common Crawl, which may be too aggressive. We also note that OpenWebMath is an English-only dataset, which limits its applications for researchers and users who speak other languages. Finally, we note that OpenWebMath only contains the *text* from math on the web, not associated figures, which can be important for solving mathematical problems (OpenAI, 2023). Future work should focus on finding empirical answers to the questions of what constitutes good data, creating new, efficient filtering methodologies, and extracting images inline with math text.

## B    OVERLAP WITH COMMON TEST SETS

We measured the n-gram overlap between OpenWebMath and a set of common benchmarks for language model reasoning. We use similar methodology to that introduced in GPT-3 (Brown et al., 2020). We used 30-gram overlap since 20-gram had a large amount of false positives due to common mathematical formulas and sequences. Table 6 shows the overlap statistics for the MATH (Hendrycks et al., 2021), GSM8k (Cobbe et al., 2021), and LILA (Mishra et al., 2022) test sets. We find that there is very little overlap between any of these benchmarks. Due to observed false positives during the overlap analysis, we opt to not perform any decontamination on OpenWebMath and leave memorization analysis to those who training models on it.

## C    DOWNSTREAM PERFORMANCE

| Benchmark | The Pile (14.7B tokens) | ProofPile (14.7B tokens) | OpenWebMath (14.7B tokens) | Mixture (14.7B tokens) | The Pile (300B tokens) |
|---|---|---|---|---|---|
| LILA-multiarith (Roy et al., 2015) | 9.77% | 8.04% | **16.67%** | 13.22% | **21.80%** |
| LILA-mathqa-probability (Amini et al., 2019) | 0.00% | 0.00% | 0.00% | 0.00% | **4.17%** |
| LILA-mathqa-general (Amini et al., 2019) | 0.13% | 0.13% | **0.51%** | 0.39% | 0.00% |
| LILA-mathqa-gain (Amini et al., 2019) | **1.02%** | **1.02%** | 0.26% | 0.26% | 0.26% |
| LILA-mathqa-geometry (Amini et al., 2019) | 0.85% | 0.85% | **2.56%** | 0.85% | **2.56%** |
| LILA-mathqa-other (Amini et al., 2019) | 0.00% | 0.00% | **1.10%** | 0.00% | 0.00% |
| LILA-mathqa-physics (Amini et al., 2019) | 0.61% | 0.82% | **1.23%** | 1.02% | **1.43%** |
| LILA-GSM8k-structured (Cobbe et al., 2021) | 1.30% | 0.84% | **2.21%** | 1.68% | 0.92% |
| LILA-add-sub (Hosseini et al., 2014) | 33.94% | 26.61% | 44.95% | **46.79%** | **53.21%** |
| LILA-asdiv (Miao et al., 2020) | 5.50% | 4.37% | 16.50% | **17.80%** | **19.58%** |
| LILA-svamp-structured (Patel et al., 2021) | 5.35% | 7.69% | 10.70% | **14.72%** | 11.71% |

Table 7: Accuracy on LILA benchmarks with numerical answers.

We tested our models trained in section 4 on a variety of downstream benchmarks to evaluate their mathematical proficiency. GSM8k (Cobbe et al., 2021) is a popular math benchmark that contains

| Method | Runtime (s) | Source Code Link |
|---|---|---|
| Resiliparse | 3.99 | https://github.com/chatnoir-eu/chatnoir-resiliparse |
| HTML-Text | 10.75 | https://github.com/TeamHG-Memex/html-text |
| Inscripts | 19.14 | https://github.com/weblyzard/inscriptis |
| BoilerPy | 24.94 | https://github.com/jmriebold/BoilerPy3 |
| jusText | 31.17 | https://github.com/miso-belica/jusText |
| HTML2Text | 37.17 | https://github.com/Alir3z4/html2text/ |
| BeautifulSoup | 38.42 | https://code.launchpad.net/beautifulsoup |
| Trafilatura | 63.90 | https://github.com/adbar/trafilatura |
| ExtractNet | 299.67 | https://github.com/currentslab/extractnet |

Table 8: We measured the performance of various HTML text extraction tools on a dataset of 1k documents. Resiliparse was by far the most efficient, leading us to choose it for use in our pipeline.

grade school math problems that are intended to be solvable by a bright middle school student. We also evaluate our models on the MATH dataset (Hendrycks et al., 2021), which contains challenging mathematics problems in informal language. Finally, we evaluated on several benchmarks which have numerical answers from the LILA (Mishra et al., 2022) evaluation suite. LILA is a collection of tasks where models are tasked to write Python code to solve word problems. These problems test the ability to do mathematical problem solving while offloading arithmetic and computation to the Python interpreter.

We evaluated GSM8k and MATH by measuring the perplexity of included chain-of-thought answers in Table 4. We evaluate the accuracy of our models on the MATH Algebra-Easy subset of the MATH benchmark, which contains only Algebra questions with a difficulty rating of level 1. We also evaluated on the full suite of LILA tasks which have numerical answers, making it simple to automatically evaluate the performance of our models. The task accuracies of our models can be found in Table 5 and Table 7. Note that models of this scale only get a few questions right on several of these benchmarks, leading to high variance.

## D  TEXT EXTRACTION

**Choice of Base Text Extractor**  When considering which HTML text-extraction library to use, we considered the efficiency, customization, and existing boilerplate reduction methods for each option. The most commonly used option, using WET files extracted by Common Crawl, was not an option since they do not deal with LaTeX correctly and offer no customization. Other options such as jusText (Endrédy & Novák, 2013), used in The Pile Gao et al. (2020), removed boilerplate too aggressively, leading to sections containing math to be discarded. Likewise, Trafilatura (Barbaresi, 2021), which was used in RefinedWeb (Penedo et al., 2023), had poor efficiency. We decided to go with Resiliparse (Bevendorff et al., 2018) due to its balanced boilerplate removal, fast runtime, and efficient Common Crawl parsing tools. Table 8 shows the full results for our comparison.

**LaTeX Extraction**  LaTeX code comes in many forms throughout Common Crawl HTML files. We employed an iterative process to refine our extraction rules. First, we filtered shards of Common Crawl for documents that contain the string `\frac`. Then, we filtered those documents to find those which our extraction code found no extractable LaTeX. Then, we refined our code to include additional sources of math until we were confident that we had reasonable support for all formats of LaTeX in HTML documents. Table 9 shows the breakdown of different common types of LaTeX found in HTML documents.

We note that most of the LaTeX in OpenWebMath and across the internet is encoded using MathJax, which presents a challenge. The majority of MathJax documents use dollar sign delimiters, but most dollar signs on the web do not delimit LaTeX equations. This leaves us with a few options:

- Detect the use of the MathJax script in the HTML file. If the script is imported, treat dollar signs as LaTeX code.

| Math Format | Percentage of Documents |
|---|---|
| Found at least one instance of math | 91.42% |
| MathJax with delimiters (inline) | 50.27% |
| MathJax with delimiters (display) | 23.37% |
| Math found in images | 6.96% |
| `.math-container` | 3.94% |
| MathML code | 3.28% |
| `<annotation>` withing `<math>` tags | 2.35% |
| `<mathjax>` tags | 2.24% |
| `align` environments | 1.72% |
| `equation` environments | 1.18% |
| within `<script>` tags | 1.01% |
| `alttext` property of `<math>` tags | 0.24% |

Table 9: Frequencies of different types of LaTeX found in OpenWebMath. The most common format of LaTeX found in Common Crawl is MathJax, which uses user-defined delimiters to denote math equations. Second most common is LaTeX code within either the URL or `alt` text of an `img` tag.

| Model Size | Layers | Model Dim | Heads | Learning Rate | Batch Size |
|---|---|---|---|---|---|
| 1.4 B | 24 | 2048 | 16 | $2.0 \times 10^{-4}$ | 1M |

Table 10: Model Hyperparameters. We use the same architecture and hyperparameters, other than batch size, as Pythia 1.4B (Biderman et al., 2023).

Table 11: List of Math Keywords used in the prefiltering stage.

**Math Keywords**

```
MathJax
mathjax
<math
math-container
katex.min.css
latex.php
codecogs
tex.cgi
class="tex"
class='tex'
```

- Detect common LaTeX commands in between dollar signs. If they are present, treat dollar signs as LaTeX code.

- Use the *MathScore* classifier to determine whether the page looks like it is talking about math. If so, treat dollar signs as LaTeX code.

The first option is not always accurate since the MathJax javascript code may be nested inside of another import or named differently depending on the website. The latter two options make up for many of these cases, but can fail to detect edge cases where math equations are present but the surrounding text does not indicate that the document is mathematical. We suspect Minerva (Lewkowycz et al., 2022) gets around this issue by using HTML documents where javascript code has already been executed, in which case MathJax is converted from delimited text to explicit HTML tags that are easy to detect.

## E    INTERPLAY BETWEEN EXTRACTION AND FILTERING

In prior works, we noticed many cases where suboptimal HTML text extractors were used and yet text quality remains high in the dataset. This is due to the interplay between extraction and filtering. Specifically, if a text extractor fails to extract the main text, gets the formatting wrong, or includes too much boilerplate in the extraction, then both the classification and perplexity filters can filter out such examples. This can lead to subtle biases in the dataset, where specific poorly-extracted websites are excluded entirely even though they do contain high quality content. In the case of making a mathematical dataset, failure to extract and deal with inline LaTeX code properly can hurt perplexity scores and lead to these documents being filtered out. We suggest practitioners tune their text extraction pipeline on a diverse set of documents before applying filtering to avoid this bias.

## F    MODEL HYPERPARAMETERS

We trained models on 14.7B tokens using the LLaMA (Touvron et al., 2023c) tokenizer and the architecture described in Pythia (Biderman et al., 2023). We train the model using the GPT-NeoX library (Andonian et al., 2023) on 8 A100 80GB GPUs. Exact hyperparameters can be found in Table 10.

## G    EXAMPLE DOCUMENT FROM OPENWEBMATH

```
Finite and Infinite-Dimensional Vector Spaces Examples 1

# Finite and Infinite-Dimensional Vector Spaces Examples 1

Recall from the Finite and Infinite-Dimensional Vector Spaces page that a
vector space $V$ is said to be finite-dimensional if there exists a set of
vectors $\{ v_1, v_2, ..., v_n \}$ in $V$ that spans $V$, that is $V = \
mathrm{span} (v_1, v_2, ..., v_n)$. If the vector space $V$ cannot be
spanned by a finite set of vectors from $V$, then $V$ is said to be infinite
-dimensional.

We will now look at some examples regarding finite and infinite-dimensional
vector spaces.

## Example 1

Show that $\wp_2 (\mathbb{R})$ is a finite-dimensional vector space by
finding a set of three polynomials $\{ p_0(x), p_1(x), p_2(x) \}$ that spans
 $\wp_2 (\mathbb{R})$. Can $\wp_2(\mathbb{R})$ be spanned by a set of two
polynomials? Can $\wp_2(\mathbb{R})$ be spanned by a set of four polynomials
?

Let $p(x) = a_0 + a_1x + a_2x^2 \in \wp_2 (\mathbb{R})$. We need to find
polynomials $p_0(x), p_1(x), p_2(x) \in \wp_2(\mathbb{R})$ such that for
scalars $b_0, b_1, b_2 \in \mathbb{R}$ we have that $p(x)$ is a linear
combination of the polynomials in $\{ p_0(x), p_1(x), p_2(x) \}$, that is:

(1)
\begin{align} \quad a_0 + a_1x + a_2x^2 = b_0p_0(x) + b_1p_1(x) + b_2p_2(x)
\end{align}

Note that if we let $p_0(x) = 1$, $p_1(x) = x$, and $p_2(x) = x^2$ then:

(2)
\begin{align} \quad a_0 + a_1x + a_2x^2 = b_0 + b_1x + b_2x^2 \end{align}

Thus for any polynomial $p(x) = a_0 + a_1x + a_2x^2 \in \wp_2 (\mathbb{R})$
we have that $b_0 = a_0$, $b_1 = a_1$, and $b_2 = a_2$. Thus $\{ 1, x, x^2
\}$ spans $\wp_2 (\mathbb{R})$.

Now there does not exist any set of two polynomials $\{ q_0(x), q_1(x) \}$
where $q_0(x) = e_0 + e_1x + e_2x^2$ and $q_1(x) = f_0 + f_1x + f_2x^2$ that
 spans $\wp_2 (\mathbb{R})$. To show this, let $c_0, c_1 \in \mathbb{R}$.
Then we'd have that:

(3)
\begin{align} \quad a_0 + a_1x + a_2x^2 = c_0q_0(x) + c_1q_1(x) \\ \quad a_0
 + a_1x + a_2x^2 = c_0(e_0 + e_1x + e_2x^2) + c_1(f_0 + f_1x + f_2x^2) \\ \
quad a_0 + a_1x + a_2x^2 = c_0e_0 + c_1f_0 + (c_0e_1 + c_1f_1)x + (c_0e_2 +
c_1f_2)x^2 \end{align}

[...]
```

Figure 5: An example of a document from OpenWebMath. Documents in OpenWebMath use Markdown formatting for elements like headings, tables, and lists, and preserve equation formatting in LaTeX. Original source: http://mathonline.wikidot.com/finite-and-infinite-dimensional-vector-spaces-examples-1

# H DATASHEET

We provide a datasheet for OpenWebMath, following the framework in Gebru et al. (2021).

| MOTIVATION | |
|---|---|
| **For what purpose was the dataset created?** | The dataset was created to enable the training of large language models on mathematical texts, in order to improve their mathematical reasoning capabilities. |
| **Who created the dataset and on behalf of which entity?** | The dataset was created by the authors of this work. |
| **Who funded the creation of the dataset?** | Resources used in preparing this research were provided, in part, by the Province of Ontario, the Government of Canada through CIFAR, Fujitsu Limited, and companies sponsoring the Vector Institute for Artificial Intelligence (`www.vectorinstitute.ai/partners`). Computing resources for model training were provided by EleutherAI and Brigham Young University. |
| **Any other comment?** | None. |
| COMPOSITION | |
| **What do the instances that comprise the dataset represent?** | The instances are text documents extracted from mathematics-related webpages from Common Crawl. |
| **How many instances are there in total?** | In total, OpenWebMath contains 6.3 million documents. |
| **Does the dataset contain all possible instances or is it a sample (not necessarily random) of instances from a larger set?** | OpenWebMath doesn't contain all instances of text extracted from mathematics-related webpages from Common Crawl, as our filters can miss a non-zero proportion of such webpages. However, we expect OpenWebMath to contain most of them. |
| **What data does each instance consist of?** | Each instance consists of plain text and metadata including the source URL, the snapshot date, and other extraction parameters. |
| **Is there a label or target associated with each instance?** | No. |
| **Is any information missing from individual instances?** | No. |
| **Are relationships between individual instances made explicit?** | No. |
| **Are there recommended data splits?** | No. |
| **Are there any errors, sources of noise, or redundancies in the dataset?** | Yes, a small portion of the documents from OpenWebMath are not related to mathematics, or contain bad quality content. |

| | |
|---|---|
| **Is the dataset self-contained, or does it link to or otherwise rely on external resources?** | The dataset is entirely self-contained. |
| **Does the dataset contain data that might be considered confidential?** | No. |
| **Does the dataset contain data that, if viewed directly, might be offensive, insulting, threatening, or might otherwise cause anxiety?** | The data is filtered for quality and we do not expect that this content will be offensive, but since our filters may be imperfect we make no guarantees. |

**COLLECTION**

| | |
|---|---|
| **How was the data associated with each instance acquired?** | The data was acquired by processing data from Common Crawl. |
| **What mechanisms or procedures were used to collect the data?** | We refer to the CommonCrawl website (commoncrawl.org) for details on how they collect data. |
| **If the dataset is a sample from a larger set, what was the sampling strategy?** | We use all data from Common Crawl that was available before May 2023. |
| **Who was involved in the data collection process and how were they compensated?** | Keiran Paster and Marco Dos Santos collected the data and were compensated by their respective graduate programs. |
| **Over what timeframe was the data collected?** | OpenWebMath uses shards of CommonCrawl gathered between 2013 and 2023. |
| **Were any ethical review processes conducted?** | No. |

**PREPROCESSING**

| | |
|---|---|
| **Was any preprocessing/cleaning/labeling of the data done?** | Yes. See section 3.5 for details. |
| **Was the "raw" data saved in addition to the preprocessed/cleaned/labeled data?** | Yes. |
| **Is the software that was used to preprocess/clean/label the data available?** | Yes. See supplementary materials. |

**USES**

| | |
|---|---|
| **Has the dataset been used for any tasks already?** | Yes, the data was used to train 1.4B parameter language models in section 4 |
| **Is there a repository that links to any or all papers or systems that use the dataset?** | No. |
| **What (other) tasks could the dataset be used for?** | We primarily envision that OpenWebMath could be useful for language model pretraining, finetuning, and evaluation. |
| **Is there anything about the composition of the dataset or the way it was collected and preprocessed/cleaned/labeled that might impact future uses?** | It is possible that the filtering stage of the project discarded valuable documents, such as those not written in English. This makes OpenWebMath suboptimal for creating mathematical models in other languages. |
| **Are there tasks for which the dataset should not be used?** | Any tasks which may considered irresponsible or harmful. |

**DISTRIBUTION**

| | |
|---|---|
| **Will the dataset be distributed to third parties outside of the entity on behalf of which the dataset was created?** | Yes, the dataset will be available on the Hugging Face Hub for NLP practitioners. |
| **How will the dataset will be distributed?** | We will distribute the dataset on the Hugging Face Hub |
| **When will the dataset be distributed?** | The dataset will be available when the paper is made public. |
| **Will the dataset be distributed under a copyright or other intellectual property (IP) license, and/or under applicable terms of use (ToU)?** | The public extract is made available under an ODC-By 1.0 license; users should also abide to the CommonCrawl ToU: https://commoncrawl.org/terms-of-use/. |
| **Have any third parties imposed IP-based or other restrictions on the data associated with the instances?** | Not to our knowledge. |
| **Do any export controls or other regulatory restrictions apply to the dataset or to individual instances?** | Not to our knowledge. |
| MAINTENANCE | |
| **Who will be supporting/hosting/maintaining the dataset?** | The dataset will be hosted on the Hugging Face Hub. |
| **How can the owner/curator/manager of the dataset be contacted?** | `keirp@cs.toronto.edu` |
| **Is there an erratum?** | No. |
| **Will the dataset be updated?** | No. |
| **If others want to extend/augment/build on/contribute to the dataset, is there a mechanism for them to do so?** | No. |

Table 12: **Datasheet for OpenWebMath**, following the framework introduced by Gebru et al. (2021).

