# OpenReview forum: "OpenWebMath: An Open Dataset of High-Quality Mathematical Web Text"
_ICLR.cc/2024/Conference — ICLR 2024 poster_

### Official Review · Reviewer_aKVX · 2023-10-23

**Soundness:** 3 good
**Presentation:** 3 good
**Contribution:** 3 good
**Rating:** 5
**Confidence:** 3

**Summary:**

In this paper, the author introduces OpenWebMath, a new large-scale dataset for language model mathematical problem-solving. A comprehensive illustration of the dataset construction pipeline is provided and some further analysis of the dataset is conducted.

**Strengths:**

1. The paper provides an opensource large-scale mathematical web text dataset which can benefit the following research.
2. The detailed dataset construction pipeline is provided.

**Weaknesses:**

1. The advance of OpenWebMath compared with existing datasets such as Proof-Pile is not provided.
2. My main concern here is that the paper is a dataset construction paper without novel technique contribution provided. I’m not very sure if this kind of paper is suitable for ICLR.

**Questions:**

What is the advance of OpenWebMath compared with existing datasets such as Proof-Pile?

---

> ### Author Response · Authors · 2023-11-19
> **Response to Reviewer aKVX**
>
> Thank you for your review and comments.
>
> > My main concern here is that the paper is a dataset construction paper without novel technique contribution provided. I’m not very sure if this kind of paper is suitable for ICLR.
> >
>
> Large language models are state-of-the-art in a large number of natural language and machine learning applications and data is often just as important or more important than other parts of the LLM training pipeline. However, data is famously one of the most closely guarded secrets of the closed labs which create these strong models. We strongly believe that dataset works such as OpenWebMath are crucial for the field’s ability to do good science in this emerging area of research.
>
> Towards the goal of pushing forward the academic study of LLM reasoning, we document as much as possible and present novel techniques throughout our pipeline to deal with the unique challenges posed by the creation of such a dataset. These include but are not limited to our methodology for extracting LaTeX code from Common Crawl files and our self-supervised filtering methods. We believe these contributions along with our analysis and openly available artifacts (filtering models, dataset) make OpenWebMath a valuable contribution to the ICLR community, especially for those who study LLM reasoning or dataset creation in the future.
>
> > The advance of OpenWebMath compared with existing datasets such as Proof-Pile is not provided.
> >
>
> Proof Pile is a hand-curated mathematics dataset that mostly ArXiv math papers. OpenWebMath is a web-scale dataset of 14.7B tokens which contains over 100k unique domains. The sources present in Proof Pile are distributed as follows:
>
> - ArXiv.math (13B tokens)
> - Open-source math textbooks and formal mathematics libraries (265M tokens)
> - Math Overflow and Math Stack Exchange (825M tokens)
> - Wiki-style sources (17M tokens)
> - MATH training set (2M tokens)
>
> Unlike OpenWebMath, Proof Pile itself is not a great dataset for math problem solving, likely since the vast majority of its data is arXiv papers which do not contain as many instances of step-by-step problem solving and usually skip basic concepts. OpenWebMath is complementary to Proof Pile and includes almost exclusively data that wasn’t in that dataset. As shown in our experiments, training on a diverse mixture of documents from OpenWebMath and Proof Pile is essential to train the best model.
>
> We will add a more thorough description of Proof Pile to the paper as well as the following table that illustrates how the two datasets are complementary.
>
> |  | Papers | Formal Math | Web Pages | Openly Available |
> | --- | --- | --- | --- | --- |
> | Minerva Web Math Pages | 21B | 0 | 17.5B | No |
> | ProofPile | 13B | 0.2B | 0.8B | Yes |
> | OpenWebMath | 0 | 0 | 14.7B | Yes |
>
> Thank you again for your review. We will update the pdf with the new paper version in the next few days. We hope that our clarifications are useful and that you will consider raising your score.

---

### Official Review · Reviewer_2Z7u · 2023-10-31

**Soundness:** 3 good
**Presentation:** 2 fair
**Contribution:** 2 fair
**Rating:** 5
**Confidence:** 3

**Summary:**

This paper proposes OpenWebMath, an open dataset of 14.7B high-quality mathematical documents from web text. The authors highlight the importance of pretraining on mathematical content to improve the reasoning abilities of large language models. They mention the success of the Minerva model, which was trained on a curated dataset of mathematical documents. However, existing open-source web datasets do not preserve mathematical notation accurately, limiting their usefulness. OpenWebMath aims to address this gap by providing a dataset of 14.7 billion tokens of mathematical web pages extracted from Common Crawl. The authors describe their method for extracting and filtering web pages for high-quality English mathematical documents. They also conduct experiments showing that models trained on OpenWebMath outperform models trained on larger general language datasets.

**Strengths:**

1. This paper proposes an open high-quality mathematical dataset, which can let models get a good reasoning ability in a lower computation.
2. This paper proposes a new method for extracting and filtering mathematical text from web pages. This method is worth deeper research.

**Weaknesses:**

1. The authors should provide an example of a dataset in the paper.
2. The order in which the table appears is inconsistent with the logic of the text.
3. There are invisible Unicode characters and some text in other languages in the data.

**Questions:**

Why there are some invisible Unicode characters and other language text in sample_dataset.jsonl? For instance the  40th and 43th lines of sample_dataset.jsonl.

---

> ### Author Response · Authors · 2023-11-19
> **Response to Reviewer 2Z7u**
>
> Thank you for your review and comments. We are happy that you find the dataset high quality and we are excited both about the models that this can be used to train and the scientific research that well documented, open datasets can enable.
>
> > The authors should provide an example of a dataset in the paper.
> >
>
> Thank you for the suggestion. We will add a figure that shows an example from the dataset to the paper.
>
> > The order in which the table appears is inconsistent with the logic of the text.
> >
>
> We will rearrange the figures in the paper to be closer to the text discussing them.
>
> > There are invisible Unicode characters and some text in other languages in the data.
> >
>
> Thank you for pointing this out. We did not do any processing to remove invisible Unicode characters simply because we were not aware this was a problem. We will add a script in our final release to remove such characters from the dataset.
>
> > Why there are some invisible Unicode characters and other language text in sample_dataset.jsonl?
> >
>
> In general, we cannot guarantee that every document in OpenWebMath is high quality since there are many documents and our filters, no matter how much they are tuned, will never be 100% accurate. Additionally, we expect a small amount of low-quality documents or a small amount of documents in another language to be present in our dataset by design, since we focus on optimizing for recall over extremely precise filtering in order to preserve more information. Users can always further filter and remove documents from OpenWebMath if their application calls for it.
>
> Still, we find that our dataset quality is often higher (by inspection) than that of other web datasets like C4 and RefinedWeb and we provide evidence in the paper that training on our dataset results in strong performance.
>
> Thank you again for your review. We will update the pdf with the new paper version in the next few days. We hope that our clarifications are useful and that you will consider raising your score.

---

> > ### Comment · Reviewer_2Z7u · 2023-11-23
> >
> > Thanks for your reply.

---

### Official Review · Reviewer_oFxt · 2023-10-31

**Soundness:** 2 fair
**Presentation:** 3 good
**Contribution:** 2 fair
**Rating:** 8
**Confidence:** 4

**Summary:**

The paper presents a large scale dataset of mathematical text (14.7B tokens, 6.3M documents) filtered from the Common Crawl dataset: OpenWebMath. The paper primarily describes the extensive pre-processing applied to obtain this dataset. To indicate the value of the dataset the paper trains a 1.4B Pythia model on the gathered data and reports perplexity on GSM8k and MATH datasets and task accuracy on MATH and LILA-multiarch. The results indicate that a model trained on OpenWebMath sees improved perplexity and better task accuracy.

**Strengths:**

- The paper presents a well documented dataset.
- The dataset seems to be useful for training LLMs of small-medium scale.

**Weaknesses:**

- The paper presents no special insight on the dataset or the effect of the processing steps applied - it only describes the pre-processing pipeline. A key aspect that would strengthen this paper is a description of the overlap (computed in some apt way: eg overlap of urls, text overlap, others) between OpenWebMath and the benchmark datasets it evaluates on - computing overlap other popular reasoning benchmarks would also be a welcome addition.

**Questions:**

- It seems like the MATH dataset was gathered from aops.com/community/c3158_usa_contests. Is this a part of Common Crawl? What is the overlap between OpenWebMath and MATH? This is important given concerns of dataset contamination with web scale datasets: https://arxiv.org/abs/2310.10628
- How does OpenWebMath differ from ProofPile? Are there obvious reasons why using OpenWebMath results in significantly better performance than ProofPile?
	- The citation for ProofPile ("Proofnet: Autoformalizing and formally proving undergraduate-level mathematics.") seems incorrect. Please consider adding a note of what the dataset is and its source.
- What exactly is LILA-multiarith? It seems the LILA benchmark contains multiple different datasets, why did the evaluation here only use this one dataset in the benchmark?
	- In similar vein to the above comments, does the data here overlap with OpenWebMath?
	- Please consider citing the original source of the multiarith dataset in addition to the benchmark, its not clear what the original data is. The citation chain to the original dataset seems to be: https://arxiv.org/pdf/2210.17517.pdf (LILA) -> https://arxiv.org/pdf/1608.01413.pdf (methodological paper using the data?) -> https://aclanthology.org/Q15-1001.pdf (original data) - please verify this.
- Please consider describing the tasks of Table 2 in more detail.
- Please place a table or figure closer to the texts discussing it.

---

> ### Author Response · Authors · 2023-11-19
> **Response to Reviewer oFxt (1/2)**
>
> Thank you for your review and for your comments. We are happy you find our dataset well-documented and useful!
>
> > The paper presents no special insight on the dataset or the effect of the processing steps applied - it only describes the pre-processing pipeline. A key aspect that would strengthen this paper is a description of the overlap (computed in some apt way: eg overlap of urls, text overlap, others) between OpenWebMath and the benchmark datasets it evaluates on - computing overlap other popular reasoning benchmarks would also be a welcome addition.
> >
>
> We believe that releasing open, well documented datasets such as OpenWebMath are critical to enabling good science in the fields of LLMs. Certainly one of the core reasons for this is that it enables greater transparency into the relationship between data overlap and downstream performance. Per your request, we evaluated the n-gram overlap between two datasets and used this to measure the overlap between our benchmarks datasets and OpenWebMath. As shown in the table below, only 348 out of 5000 MATH problems, 2 out of 1319 GSM8k problems, and 2 of 1310 LILA problems tested overlap with OpenWebMath. Due to the potential for false positives that we discovered during our overlap analysis, we opt to leave decontamination to those training models on the dataset. We will update the paper to include this overlap analysis (and include the code and overlap locations in our release) and we are open to adding other reasoning benchmarks to this analysis if they are suggested.
>
> | Evaluation | Problems with 30-gram Overlap | Total Problems |
> | --- | --- | --- |
> | MATH | 348 | 5000 |
> | GSM8k | 2 | 1319 |
> | LILA (all tested) | 3 | 1310 |
>
> > The citation for ProofPile ("Proofnet: Autoformalizing and formally proving undergraduate-level mathematics.") seems incorrect. Please consider adding a note of what the dataset is and its source.
> >
>
> Proof Pile is hosted here: https://huggingface.co/datasets/hoskinson-center/proof-pile. We have checked with the authors of Proof Pile and confirmed that the Proofnet citation is correct.
>
> > How does OpenWebMath differ from ProofPile? Are there obvious reasons why using OpenWebMath results in significantly better performance than ProofPile?
> >
>
> Proof Pile is a hand-curated mathematics dataset that mostly ArXiv math papers. OpenWebMath is a web-scale dataset of 14.7B tokens which contains over 100k unique domains. The sources present in Proof Pile are distributed as follows:
>
> - ArXiv.math (13B tokens)
> - Open-source math textbooks and formal mathematics libraries (265M tokens)
> - Math Overflow and Math Stack Exchange (825M tokens)
> - Wiki-style sources (17M tokens)
> - MATH training set (2M tokens)
>
> Unlike OpenWebMath, Proof Pile itself is not a great dataset for math problem solving, likely since the vast majority of its data is arXiv papers which do not contain as many instances of step-by-step problem solving and usually skip basic concepts. OpenWebMath is complementary to Proof Pile and includes almost exclusively data that wasn’t in that dataset. As shown in our experiments, training on a diverse mixture of documents from OpenWebMath and Proof Pile is essential to train the best model.
>
> We will add a more thorough description of Proof Pile to the paper as well as the following table that illustrates how the two datasets are complementary.
>
> |  | Papers | Formal Math | Web Pages | Openly Available |
> | --- | --- | --- | --- | --- |
> | Minerva Web Math Pages | 21B | 0 | 17.5B | No |
> | ProofPile | 13B | 0.2B | 0.8B | Yes |
> | OpenWebMath | 0 | 0 | 14.7B | Yes |

---

> ### Author Response · Authors · 2023-11-19
> **Response to Reviewer oFxt (2/2)**
>
> > What exactly is LILA-multiarith? It seems the LILA benchmark contains multiple different datasets, why did the evaluation here only use this one dataset in the benchmark?
> >
>
> LILA is a large collection of math benchmarks of varying quality and required skill levels, so we just picked one that we thought tested a unique aspect relative to our existing benchmarks. The LILA evaluations involve running Python code in order to produce an answer and LILA-multiarith is essentially a Python version of the multiarith benchmark.
>
> We have now evaluated our models on all the LILA evaluations which have numerical answers and automatic evaluation below. Note the accuracies are quite low on some of the evaluations across the board due to their difficulty, which can result in a lot of variance in scores - we believe perplexity evaluations (included in the paper) are the most appropriate for measuring the strength of datasets like OpenWebMath at this type of scale since performance scales more smoothly with this metric [1].
>
> |  | The Pile | ProofPile | OpenWebMath | Mixture | Pythia 1.4b |
> | --- | --- | --- | --- | --- | --- |
> | MATH-Algebra-Easy | 2.81% | 2.81% | **5.62%** | 5.06% | 3.93% |
> | MATH-Algebra-Easy maj@16 | 3.93% | 3.93% | 9.55% | **10.11%** | 5.62% |
> | LILA-multiarith | 9.77% | 8.04% | **16.67%** | 13.22% | **21.80%** |
> | LILA-mathqa-probability | 0.00% | 0.00% | 0.00% | 0.00% | **4.17%** |
> | LILA-mathqa-general | 0.13% | 0.13% | **0.51%** | 0.39% | 0.00% |
> | LILA-mathqa-gain | **1.02%** | **1.02%** | 0.26% | 0.26% | 0.26% |
> | LILA-mathqa-geometry | 0.85% | 0.85% | **2.56%** | 0.85% | **2.56%** |
> | LILA-mathqa-other | 0.00% | 0.00% | **1.10%** | 0.00% | 0.00% |
> | LILA-mathqa-physics | 0.61% | 0.82% | **1.23%** | 1.02% | **1.43%** |
> | LILA-GSM8k-structured | 1.30% | 0.84% | **2.21%** | 1.68% | 0.92% |
> | LILA-add-sub | 33.94% | 26.61% | 44.95% | **46.79%** | **53.21%** |
> | LILA-asdiv | 5.50% | 4.37% | 16.50% | **17.80%** | **19.58%** |
> | LILA-svamp-structured | 5.35% | 7.69% | 10.70% | **14.72%** | **11.71%** |
>
> > In similar vein to the above comments, does the data here overlap with OpenWebMath?
> >
>
> Please see our above response for the data overlap results for LILA.
>
> > Please consider citing the original source of the multiarith dataset in addition to the benchmark, its not clear what the original data is. The citation chain to the original dataset seems to be: https://arxiv.org/pdf/2210.17517.pdf (LILA) -> https://arxiv.org/pdf/1608.01413.pdf (methodological paper using the data?) -> https://aclanthology.org/Q15-1001.pdf (original data) - please verify this.
> >
>
> We will clean up our citations in order to cite the original sources of the LILA benchmarks.
>
> > Please consider describing the tasks of Table 2 in more detail.
> >
>
> MATH Algebra-Easy is the Algebra subset of the MATH benchmark, filtered down to only questions with difficulty level 1. We evaluated our models on MATH Algebra-Easy with both greedy decoding and self-consistency. We also evaluated models on LILA tasks, which involve writing Python code to solve a math word problem and contains many subsets corresponding to different types of math problems. These problems test the ability to do mathematical problem solving while offloading arithmetic and computation to the Python interpreter. We tested on all the LILA subsets which have numerical answers, making it simple to automatically evaluate the performance of our models. We will add a paragraph with this information to the paper.
>
> > Please place a table or figure closer to the texts discussing it.
> >
>
> We will rearrange the figures in the paper to be closer to the text discussing them.
>
> Thank you again for your review. We will update the pdf with the new paper version in the next few days. We hope that our clarifications are useful and that you will consider raising your score.
>
> [1] Schaeffer et al. "Are Emergent Abilities of Large Language Models a Mirage?" arXiv 2023.

---

> > ### Comment · Reviewer_oFxt · 2023-11-22
> > **Thank you for the clarifications.**
> >
> > Thank you for the clarifications and the updates to the paper. Is there a reason 30-gram overlap was used to measure overlap with the test set? What is the level of overlap with smaller n-grams, eg, 5, 10, 20?

---

> > > ### Author Response · Authors · 2023-11-22
> > > **Overlap Clarification (1/2)**
> > >
> > > The reason we chose 30 is because with a lower $n$, there tend to be too many false positives. We have run the overlap analysis with the MATH test set with $n=10$ and $n=20$ now and report the results in the table below.
> > >
> > > | n | MATH Test Set Problem Statements with n-gram Overlap |
> > > | --- | --- |
> > > | 10 | 1334 |
> > > | 20 | 607 |
> > > | 30 | 348 |
> > >
> > > We investigated the overlapping examples with each value of $n$ and found that the number of false positives is quite high at lower values. At $n=30$, we seem to catch mostly every case of overlap that $n=20$ catches without the false positives. We post examples of the first three overlapping examples for each value of $n$ below. As you can see, there are plenty of common phrases and equations in math questions that can easily lead to overlap.
> > >
> > > ### N = 10
> > >
> > > **Example 1:**
> > >
> > > **MATH Problem Statement**: `Determine the modulo 4 remainder of the following sum: $$ 1 + 2 + 3 + 4 + 5 + 6 + 7 + 8 + 9 + 10 + 11 + 12. $$`
> > >
> > > **Overlapping Document Snippet**: `For example, there are 17977 partitions for the number 36; it is both a square number $(6^2)$ and a triangular number $(1 + 2 + <span style="color: red;">3 + 4 + 5 + 6 + 7 +</span> 8).$ Aesthetic`
> > >
> > > **Example 2:**
> > >
> > > **MATH Problem Statement**: `The smallest distance between the origin and a point on the parabola $y=x^2-5$ can be expressed as $\sqrt{a}/b$, where $a$ and $b$ are positive integers, and $a$ is not divisible by the square of any prime. Find $a+b$.`
> > >
> > > **Overlapping Document Snippet**: `Then $h$ can be written in the form $\frac{k\sqrt{m}}{n}$, where $k$ and $n$ are relatively prime positive integers and $m$ is a positive integer that is not divisible by the square of any prime. Find $k+m+n$.`
> > >
> > > **Example 3:**
> > >
> > > **MATH Problem Statement**: `Let $P$ be the convex polygon in the complex plane whose vertices are the roots of [z^7 + z^6 + z^5 + z^4 + z^3 + z^2 + z + 1 = 0.]The area of $P$ can be expressed in the form $\frac{a + b \sqrt{c}}{d},$ where $a,$ $b,$ $c,$ $d$ are positive integers, in simplest form. Find $a + b + c + d.$`
> > >
> > > **Overlapping Document Snippet**: `sage: pari.polcyclo(8) x^4 + 1 sage: pari.polcyclo(7, 'z') z^6 + z^5 + z^4 + z^3 + z^2 + z + 1 sage: pari.polcyclo(1) x - 1`
> > >
> > > ### N = 20
> > >
> > > **Example 1:**
> > >
> > > **MATH Problem Statement:** `What number must be placed in the box in the equation below to produce an equation that has more than one solution: [4x + 6 + 7x - 9 = 12x - 7 - x + \boxed{\phantom{2}}?]`
> > >
> > > **Overlapping Document Snippet:** `What number must be placed in the box in the equation below to produce an equation that has more than one solution: $1/2*y + 1/4 = 3 + \boxed{\phantom{400} } y$`
> > >
> > > **Example 2:**
> > >
> > > **MATH Problem Statement:** `Let $A(2,2)$ and $B(7,7)$ be points in the plane. Define $R$ as the region in the first quadrant consisting of those points $C$ such that $\triangle ABC$ is an acute triangle. What is the area of the region $R$?`
> > >
> > > **Overlapping Document Snippet:** `Let $A(2,2)$ and $B(7,7)$ be points in the plane. Define $R$ as the region in the first quadrant consisting of those points $C$ such that $\triangle ABC$ is an acute triangle. What is the closest integer to the area of the region $R$?`
> > >
> > > **Example 3:**
> > >
> > > **MATH Problem Statement:** `Let $P(x)$ be a quadratic polynomial with real coefficients satisfying $x^2 - 2x + 2 \le P(x) \le 2x^2 - 4x + 3$ for all real numbers $x$, and suppose $P(11) = 181$. Find $P(16)$.`
> > >
> > > **Overlapping Document Snippet:** `Let $P(x)$ be a quadratic polynomial with real coefficients satisfying $x^2 - 2x + 2 \le P(x) \le 2x^2 - 4x + 3$ for all real numbers $x$, and suppose $P(11) = 181$. Find $P(16)$.`

---

> > > > ### Author Response · Authors · 2023-11-22
> > > > **Overlap Clarification (2/2)**
> > > >
> > > > ### N = 30
> > > >
> > > > **Example 1:**
> > > >
> > > > **MATH Problem Statement:** `Let $A(2,2)$ and $B(7,7)$ be points in the plane. Define $R$ as the region in the first quadrant consisting of those points $C$ such that $\triangle ABC$ is an acute triangle. What is the area of the region $R$?`
> > > >
> > > > **Overlapping Document Snippet:** `Let $A(2,2)$ and $B(7,7)$ be points in the plane. Define $R$ as the region in the first quadrant consisting of those points $C$ such that $\triangle ABC$ is an acute triangle. What is the closest integer to the area of the region $R$?`
> > > >
> > > > **Example 2:**
> > > >
> > > > **MATH Problem Statement:** `Let $P(x)$ be a quadratic polynomial with real coefficients satisfying $x^2 - 2x + 2 \le P(x) \le 2x^2 - 4x + 3$ for all real numbers $x$, and suppose $P(11) = 181$. Find $P(16)$.`
> > > >
> > > > **Overlapping Document Snippet:** `Let $P(x)$ be a quadratic polynomial with real coefficients satisfying $x^2 - 2x + 2 \le P(x) \le 2x^2 - 4x + 3$ for all real numbers $x$, and suppose $P(11) = 181$. Find $P(16)$.`
> > > >
> > > > **Example 3:**
> > > >
> > > > **MATH Problem Statement:** `Bob and Alice each have a bag that contains one ball of each of the colors, blue, green, orange, red, and violet. Alice randomly selects one ball from her bag and puts it into Bob's bag. Bob then randomly selects one ball from his bag and puts it into Alice's bag. What is the probability that after this process the contents of the two bags are the same?`
> > > >
> > > > **Overlapping Document Snippet:** `Bob and Alice each have a bag that contains one ball of each of the colors blue, green, orange, red, and violet. Alice randomly selects one ball from her bag and puts it into Bob's bag. Bob then randomly selects one ball from his bag and puts it into Alice's bag. What is the probability that after this process the contents of the two bags are the same?`
> > > >
> > > > We hope that our clarifications have cleared up any questions you have about our overlap analysis and that you will consider raising your score. Please let us know if you have any further questions.

---

> > > > > ### Comment · Reviewer_oFxt · 2023-11-22
> > > > > **Raised my score.**
> > > > >
> > > > > Thanks for the clarifications. I would encourage the inclusion of the above clarifications and examples in the paper's appendix to substantiate the choice of 30 grams. I have raised my score.

---

### Author Response · Authors · 2023-11-21
**General Comments to All Reviewers**

Thanks to all the reviewers for your time during the review process. We appreciate that you found our work well-document, high-quality, and beneficial to future research.

We have responded to each reviewer individually and additionally updated the PDF to address feedback.

The new version of the paper contains the following changes:

- We include an analysis of the overlap between OpenWebMath and common test sets in Appendix B.
- We have added the original citations for all of the LILA tasks we used.
- We have updated section 4 to include more information about the difference between ProofPile and OpenWebMath.
- We have added a new table containing 10 additional LILA task accuracies to Appendix C.
- We have rearranged the tables and figures in the paper to place them closer to the text they accompany.
- We added a description of our evaluation tasks in Appendix C.
- We provide an example of a document from OpenWebMath in Appendix G.

We again thank the reviewers for their feedback, which we believe has substantially improved the paper. We ask reviewers to please read through our clarifications and the updated PDF and to update their scores if we have addressed their concerns.

---

### Meta-Review · Area_Chair_EWkh · 2023-12-13

**Metareview:**

This submission introduces OpenWebMath --- An Open Dataset of High-Quality Mathematical Web Text --- for training large language models. It contains 14B tokens filtered from CC. The original submission introduces the extensive pre-processing applied to obtain this dataset. It demonstrates the dataset's value by training a relatively small-scale model (1.4B) on it and evaluating on GSM8k and MATH datasets, as well as task accuracy on MATH and LILA-multiarch. The findings show that training on OpenWebMath leads to improved task accuracy.

The submission has received mixed reviews, with scores of 8, 5, and 5. The two reviews with a score of 5 are relatively brief and indicate questions on how to align the work's contributions --- a dataset --- with the standards of ICLR.

The author(s) demonstrate the dataset's worth by training a 1.4B model. From my first-hand experience, OpenWebMath can significantly enhance the mathematical problem-solving abilities of large language models with tens to hundreds of billions of parameters. It's more useful than some highly-rated ICLR'24 submissions on the same topic, at least in our experience.

 Honestly, it is way more useful that some of high-rating ICLR'24 submissions on the same topic, at least from our own experiences. On one hand, it would be unfortunate that this valuable dataset would not be recognized and used by the community. However, the execution of the original submission indeed requires further improvements. Reviewer oFxt gives very detailed inquiries and suggestions to further improve this work and the author(s) addresses most of them by revising and updating the submission with following changes:

1. We include an analysis of the overlap between OpenWebMath and common test sets in Appendix B.
2. We have added the original citations for all of the LILA tasks we used.
3. We have updated section 4 to include more information about the difference between ProofPile and OpenWebMath.
4. We have added a new table containing 10 additional LILA task accuracies to Appendix C.
5. We have rearranged the tables and figures in the paper to place them closer to the text they accompany.
6. We added a description of our evaluation tasks in Appendix C.
7. We provide an example of a document from OpenWebMath in Appendix G.

All things considered, I would give a positive recommendation to this work given it is very difficult to identify useful datasets/techniques to improve LLM training these days due to it excessive compute cost. However, I would suggest the author(s) to incorporate more experimental results (e.g., 4) in the main paper to make it a more self-contained and strong demonstration.

**Justification For Why Not Higher Score:**

This work worths an acceptance given it provides a useful dataset for improving LLM's math problem-solving ability. However, the execution of this work requires further improvements.

**Justification For Why Not Lower Score:**

n/a

---

### Decision · Program_Chairs · 2024-01-16

Accept (poster)